# Reversed evolution of grazer resistance to cyanobacteria

Jana Isanta-Navarro[1,5 ✉], Nelson G. Hairston Jr [2], Jannik Beninde [3], Axel Meyer [3], Dietmar Straile [1], Markus Möst [4] & Dominik Martin-Creuzburg [1]

Exploring the capability of organisms to cope with human-caused environmental change is crucial for assessing the risk of extinction and biodiversity loss. We study the consequences of changing nutrient pollution for the freshwater keystone grazer, *Daphnia*, in a large lake with a well-documented history of eutrophication and oligotrophication. Experiments using decades-old genotypes resurrected from the sediment egg bank revealed that nutrient enrichment in the middle of the 20th century, resulting in the proliferation of harmful cyanobacteria, led to the rapid evolution of grazer resistance to cyanobacteria. We show here that the subsequent reduction in nutrient input, accompanied by a decrease in cyanobacteria, resulted in the re-emergence of highly susceptible *Daphnia* genotypes. Expression and subsequent loss of grazer resistance occurred at high evolutionary rates, suggesting opposing selection and that maintaining resistance was costly. We provide a rare example of reversed evolution of a fitness-relevant trait in response to relaxed selection.

[1] Limnological Institute, University of Konstanz, Konstanz, Germany. [2] Cornell University, Department of Ecology and Evolutionary Biology, Ithaca, USA. [3] Zoology and Evolutionary Biology, University of Konstanz, Konstanz, Germany. [4] University of Innsbruck, Department of Ecology, Innsbruck, Austria. [5] Present address: Flathead Lake Biological Station, University of Montana, Polson, USA. ✉email: jana.isantanavarro@flbs.umt.edu

Lake ecosystems around the globe are suffering from environmental changes that are caused or intensified by human activity, with severe consequences for lake environments, their organisms and their value to people. In the past century, urbanization, intensive agriculture and sewage discharge have been among the major threats to aquatic ecosystems, resulting in increased nutrient loads, reduced water quality, and harmful cyanobacterial blooms[1–3]. Cyanobacteria are nutritionally inadequate for zooplankton grazers, because of morphological properties, the production of toxic secondary metabolites, and the lack of essential sterols and long-chain polyunsaturated fatty acids (PUFAs)[4–7]. When cyanobacteria dominate the phytoplankton community, the freshwater keystone grazer *Daphnia* is subject to strong natural selection[8–10], expected to favor those genotypes that are able to survive, grow and reproduce on a cyanobacteria-rich diet. In Lake Constance, a large pre-Alpine lake in Central Europe, massive eutrophication (from 1950s to 1980) was accompanied by rapid evolution of *Daphnia* resistance to cyanobacteria, as revealed in growth experiments with decades-old *Daphnia* genotypes hatched ("resurrected"[11]) from dormant eggs that were chronologically deposited into the sediment[12,13]. Since then, intense restoration efforts resulted in reduced nutrient loads (oligotrophication) and a substantial decrease in cyanobacterial biomass[14,15].

Natural selection has been at the center of evolutionary studies since Wallace[16] and Darwin[17]. While the great majority of research effort has focused on understanding the adaptive evolution of traits in response to natural selection, exploring the fate of previously evolved traits following abatement of a selection pressure (relaxed selection[18]) has received much less attention. Natural populations are increasingly exposed to anthropogenic or natural environmental changes[19], some of which weaken or even remove a source of selection. However, empirical examples from natural populations experiencing relaxed selection are scarce, hampering our understanding of the patterns and mechanisms of trait decay[18,20–22]. Especially in light of current efforts to minimize and reverse human impacts on ecosystems, it is of particular importance to explore how species react to human-caused environmental change. Resurrecting *Daphnia* genotypes from the sediment egg bank of Lake Constance thus allows reconstructing evolutionary processes related to eutrophication followed by oligotrophication, including the consequences of relaxed selection for a previously evolved fitness-relevant trait.

Here, we explore experimentally how the capacity of *Daphnia* to cope with cyanobacterial food changed with oligotrophication of Lake Constance. We hypothesize that recent *Daphnia* genotypes are more susceptible to cyanobacteria than genotypes from when the lake was eutrophic, due to the decline in cyanobacteria abundance, and that the resistance to cyanobacteria was lost in the population through reversed directional selection. Life-history experiments were conducted with *Daphnia galeata* genotypes resurrected from diapausing eggs that were isolated from dated sediment layers representing peak-eutrophic (hereafter 1980s) and re-oligotrophic conditions (hereafter 2000s). We compare the obtained data with previous results showing the adaptive evolution of resistance using genotypes from pre-eutrophic (hereafter 1960s) and peak-eutrophic conditions[12,13]. This allows us to document trait changes over almost half a century and to disentangle the different ecological and evolutionary components mediating the evolution of resistance to cyanobacteria.

## Results and discussion

**Reversed evolution**. Our data demonstrate that the trait 'resistance to cyanobacteria', which underlay adaptive evolution during eutrophication[12,13], was subject to reversed selective forces during subsequent oligotrophication, i.e., after relaxation of the selection pressure. To quantify resistance to cyanobacteria, we measured the reduction in juvenile somatic growth rates of resurrected *Daphnia* genotypes upon exposure to a toxic cyanobacterium (*Microcystis aeruginosa*). Growth rate reduction (GRR) was calculated from growth rate differences on good food (100% green alga *Acutodesmus obliquus*) and poor food (20% *M. aeruginosa* and 80% *A. obliquus*; Fig. 1a). The data obtained were compared with previously established GRRs resulting from adaptive evolution during eutrophication[12,13] (Fig. 1a). Genotypes from the 1980s expressed a significantly lower GRR (=higher resistance) than genotypes from the 2000s (=lower resistance; ANOVA, $F_{1,16} = 62.6$, $p < 0.001$; Fig. 1a). Strikingly, the ranges in GRRs expressed by genotypes from the 1980s and 2000s did not overlap, indicating strong selection against resistance rather than neutral drift after relaxation of the selection pressure. Trait loss due to neutral processes is expected to proceed much more slowly than through reversed selection[23]. The rates of evolution we document here (Fig. S2) suggest that the trait 'resistance to cyanobacteria' is heritable and was subject to reversed selection, and not simply shaped by neutral processes. The intermediate GRRs expressed by genotypes from the 1990s imply a progressive increase in mean GRR, providing further evidence for reversed evolution. Combining our data with earlier data from Hairston et al.[12,13] offers a clear perspective on a natural population that evolved along two sequential evolutionary trajectories, i.e., adaptive and reversed adaptive evolution, both driven by human activities.

Reversed selection on a previously adaptive trait should occur when the relevant selection pressure is removed and maintaining the trait bears costs (fluctuating selection[24–26]). In our example, the trajectories of evolution show a reversal of selection for 'juvenile somatic growth rate' (Fig. 1c). However, reversed evolution did not end with genotypes showing trait expression equivalent to that of genotypes from the 1960s at the start of eutrophication. In fact, genotypes from the 2000s exhibited lower growth rates on cyanobacteria than genotypes from the 1960s (Fig. 1c), suggesting that genotypes from the 1960s were less sensitive to the presence of cyanobacteria. This is consistent with the finding that the cyanobacterial biomass has already started to increase in the late 1960s, i.e., at the end of the time period from which the "pre-eutrophic" genotypes were isolated (Fig. 1b). Thus, the genotypes from the 1960s had been exposed to low densities of cyanobacteria during the onset of eutrophication, potentially explaining their, on average, higher growth potential on the cyanobacteria-containing diet. Hence, the substantial changes in growth characteristics occurring in the *Daphnia* population within the past decades can be related to the level of occurrence of cyanobacteria in the lake.

*D. galeata* abundance is below detection level in the water column of Lake Constance during winter, which strongly suggests that annual recruitment from the sediment egg bank is crucial for population persistence[27]. Recruitment from the sediment egg bank typically occurs from the upper few millimeters of sediment; diapausing eggs from deeper sediment layers are lost to the population, unless the sediment is disturbed[28]. The origin of the genotypes with lower resistance to cyanobacteria, present in the lake in the 2000s, is unclear. They may have remained present in the water column in low frequency, even while the lake was eutrophic, they may have hatched from older sediment layers deposited during pre-eutrophic times (requiring sediment disturbance), or they may have been introduced by dispersal from nearby lakes. Alternatively, they may represent either new mutations (possible since the population size is very large), or introgression of genes from hybridization with a sister taxon, *Daphnia longispina*, also present in the lake.

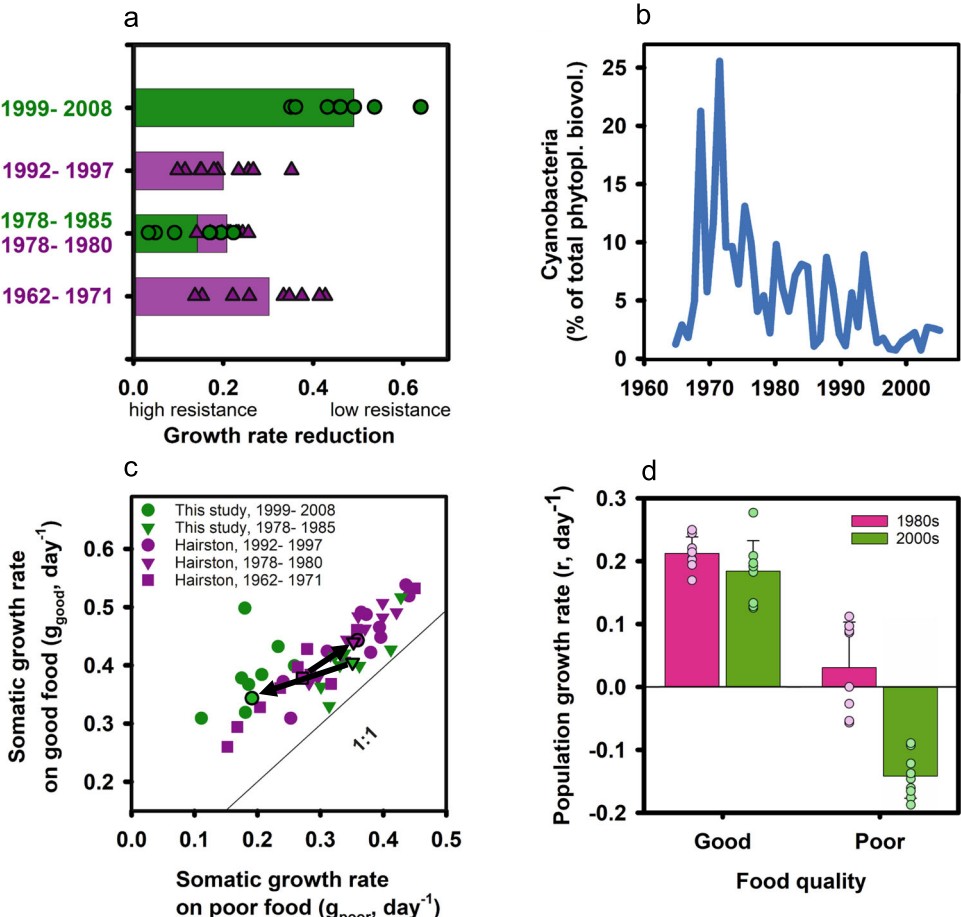

**Fig. 1 Performance of *Daphnia* genotypes and cyanobacteria biomass. a** Resistance of *Daphnia* genotypes to toxic cyanobacteria. Single data points depict average (i = 3) growth rate reduction (GRR) for individual genotypes, bars show the average GRR of all genotypes within a single time period (green dots and bars; high GRR = low resistance; one-way ANOVA, $F_{1,16} = 62.6$, $p < 0.001$). For comparison, data from Hairston et al.[12] (purple triangles and bars) are included, showing the rapid adaptive evolution of *Daphnia* genotypes to dietary cyanobacteria. **b** Contribution of cyanobacteria to total phytoplankton biovolume (blue line) in Lake Constance[43]. **c** Growth rate on good food always exceeded that on poor food. Average juvenile somatic growth rates of genotypes from five-time periods are depicted in purple (Hairston et al.[12]) and green (this study) symbols. Mean juvenile growth rates (symbols with black outline) of the different time zones are connected with black arrows, the trajectories of evolution are opposed (black arrows). Evolution of mean juvenile growth rate moved parallel to the 1:1 line from 1960s to 1990s but away from the 1:1 line from 1980 to 2000s (reversed evolution of grazer resistance during oligotrophication). **d** *Per capita* population growth rates (r) of genotypes from the 1980s (pink bars and symbols; n = 8 biologically independent genotypes) and 2000s (green bars and symbols; n = 9 biologically independent genotypes) on good and poor food. Bars represent mean values and standard deviation, single data points depict average population growth rate of one genotype. On poor food, none of the genotypes from 2000s was able to reproduce, while five out of eight genotypes from 1980s were able to produce three consecutive clutches with viable offspring. For genotypes that did not produce any offspring on poor quality food, population growth rates reflect the *per capita* death rates (see methods). Population growth rates differed significantly between the two time periods on the poor food ($p = <0.001$) but not on the good food ($p = 0.241$; Tukey's HSD following two-way ANOVA: food: $F_{1,33} = 233.8$, $p < 0.001$; time zone: $F_{1,33} = 36.5$, $p < 0.001$; food × time zone: $F_{1,33} = 18.9$, $p < 0.001$). Source data are provided as a Source Data file.

Within the two time periods 1980s and 2000s, clonal reaction norms as a function of growth on good and poor food varied significantly, indicating genetic variation in resistance to dietary cyanobacteria. The slopes of the reactions norms were significantly steeper (ANOVA, $F_{1,16} = 18.8$, $p < 0.001$; Fig. 2b) for genotypes from the 2000s than for genotypes from the 1980s. Preceding this increase in reaction norm slope from 1980s to 2000s was a decrease in reaction norm slope from the 1960s to 1980s[13] (Fig. 2a). The higher contribution of ancestral phenotypic plasticity (Fig. S3) to the observed trait change during eutrophication implies that the genotypes that invaded Lake Constance in the late 1950s, i.e., early during eutrophication, expressed high phenotypic plasticity and originated from a lake (or several lakes) without or only sporadically occurring cyanobacterial blooms and thus were sensitive to the presence of cyanobacteria in their diet. *Daphnia* evolved resistance to cyanobacteria with increasing eutrophication and

consequently, the plasticity of the trait decreased[13]. Juvenile somatic growth rates became more similar in the two food environments as resistance evolved, i.e., growth rates increased in the cyanobacteria environment, while changing little in the presence of good food (Fig. 2a). During subsequent oligotrophication, resistance to cyanobacteria was lost (Fig. 2b) and growth rates on poor food decreased markedly presumably because the costs of maintaining resistance to dietary cyanobacteria were high. This led to the evolution of increased plasticity in the good food environment (Fig. S3), demonstrating that evolution of plasticity is a critical component in the adaptive response of populations to human-caused environmental change[19].

The realized juvenile growth rates, meaning growth rates in the food environment the genotypes actually experienced in their respective time period, did not differ significantly, i.e., genotypes from 1980s (when cyanobacteria were abundant) achieved growth

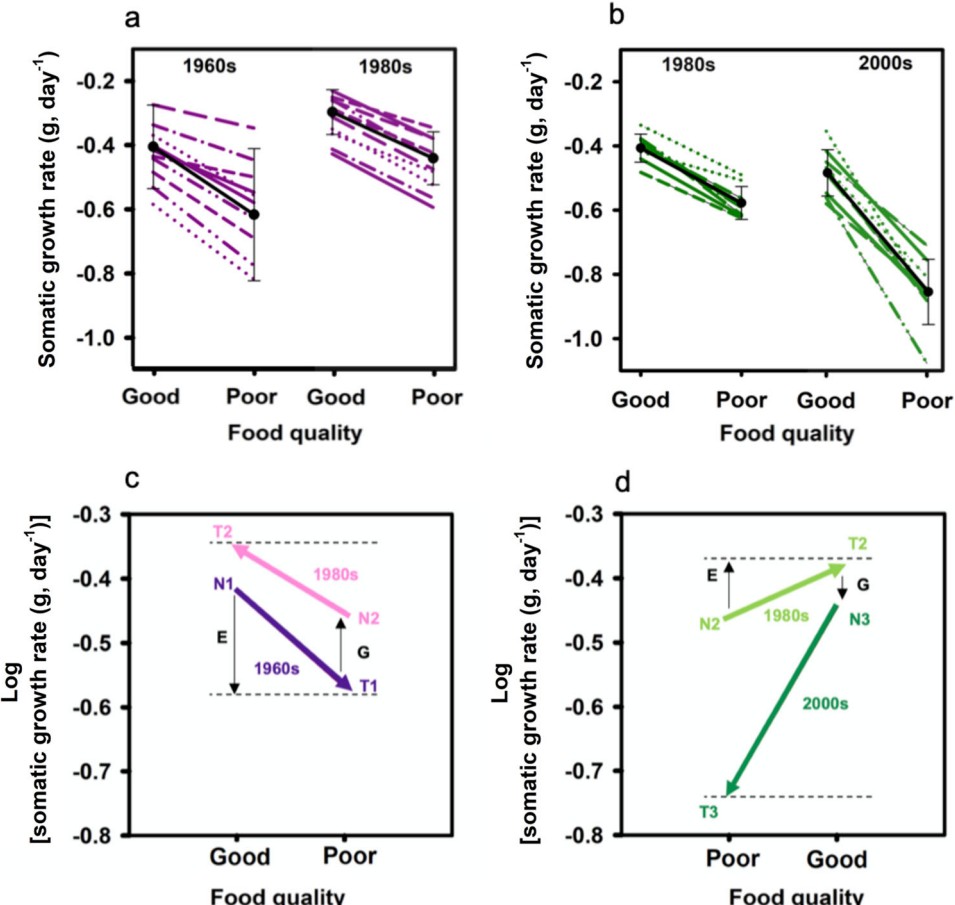

**Fig. 2 Reaction norms and counter-gradient variation.** Reaction norms of *Daphnia* (log transformed juvenile somatic growth rates) as a function of food environment (good and poor food; (**a**) data from Hairston et al.[12] (1960s $n = 10$ biologically independent genotypes; 1980s $n = 10$ biologically independent genotypes) (**b**) this study (1980s $n = 8$ biologically independent genotypes; 2000s $n = 9$ biologically independent genotypes; one-way ANOVA on slopes, $F_{1,16} = 0.22$, $p = 0.65$). Each line depicts one genetically distinct genotype (purple and green for (**a**) and (**b**), respectively). Black lines depict the mean reaction norm of genotypes from a single time period (±SD). Mean trait change (juvenile growth rate) within *D. galeata* genotypes from oligotrophic (1962–1971) and peak-eutrophic times (1978–1980). **c, d** The outcome of the common garden experiments, in which *Daphnia* genotypes from the 1960s, 1980s, and 2000s were exposed to good and poor food quality (i.e., without and with cyanobacteria). The lines represent the mean reaction norms (phenotypic plasticity) for juvenile somatic growth rates (log data) for each of the populations (purple 1960s, pink 1980s, light green 1980s, and dark green 2000s). The end points N show the mean growth rates of the populations on the food that is assumed to represent the food environment experienced in the field at a certain time period; the end points T show the mean growth rates displayed upon exposure to the alternative food environment. The vertical arrows (black) indicate the opposing effects of environmental and genetic influences. Source data are provided as a Source Data file.

rates on poor food similar to those of genotypes from 2000s (when cyanobacteria were scarce) on good food (ANOVA, $F_{1,16} = 0.22$, $p = 0.65$; Fig. 2b) and genotypes from the 1960s achieved similar growth rates on good food to genotypes from the 1980s on poor food[13]. Neonate *Daphnia* did not differ in body mass, so that juvenile somatic growth rate should be closely tied to adult body size. Thus, genotypes from peak-eutrophic and re-oligotrophic conditions presumably obtained roughly the same adult body size, suggesting that size-selective predation by zooplanktivorous fish and invertebrates did not change significantly over time or was too low to induce life-history changes[13]. In addition to juvenile somatic growth rates, we also recorded mortality, age at reproduction, and the number of offspring produced within the first three reproduction cycles, and from these estimated population growth rates (Fig. 1d). The decay of resistance to cyanobacteria is reflected in all life-history traits recorded here, indicating an absence of trade-offs among fitness-relevant traits[29]. Reproductive failure and high mortality on the cyanobacteria-containing diet suggest that the ability to cope with

dietary cyanobacteria was lost completely in the *Daphnia* population following oligotrophication.

**Genetic and environmental effects.** Using genotypes originating from 1992 to 1997, Hairston et al.[13] did not find a change in mean tolerance of the *D. galeata* population to cyanobacteria, but noticed a modest increase in variation among clonal reaction norms, pointing already towards evolutionary changes with ongoing oligotrophication. Taken together, this implies strong interactions between genetic and environmental effects (G × E) on growth rate expression during both transitions, i.e., eutrophication (adaptive evolution) and oligotrophication (reversed adaptive evolution). We show here that the genetic and environmental influences on phenotypes opposed one another during both transitions (Fig. 2c, d), thereby diminishing the change in mean trait expression across the temporal environmental gradient (here cyanobacteria abundance), consistent with the counter-gradient variation concept[30]. Counter-gradient selection is a process occurring whenever environmental influences on trait

expression push the phenotype away from its optimum in a given environment. Counter-gradient variation (CnGV) is the adaptive response that may evolve to counteract potentially detrimental effects on fitness[30]. Accordingly, the magnitude of phenotypic change across environments is determined by genetic (G) and environmental (E) influences, as well as their interaction (G × E) and covariance (Cov(G,E)). The Cov(G,E) term can either be positive or negative depending on whether genetic and environmental effects reinforce or oppose each other[31]. Our data provide evidence for CnGV (Cov(G,E) < 0) during both transitions, i.e., adaptive evolution (eutrophication) and reversed adaptive evolution (oligotrophication). The CnGV approach is commonly applied to assess local adaptation along latitudinal or altitudinal gradients but has been also used to explore phenotypic and genotypic changes on temporal scales encompassing decades of environmental change[32–34]. The resurrection ecology approach applied here has the advantage that genotypes resurrected from the past can be directly exposed to different environmental conditions, thus facilitating the assessment of environmental and genetic influences along a temporal gradient.

**Reversed selection on a costly trait**. The loss of grazer resistance to cyanobacteria with oligotrophication implies that expressing and maintaining this previously evolved trait incurred substantial fitness costs. Genotypes from the 2000s were more susceptible to the cyanobacterial diet than genotypes from the 1980s. In the absence of cyanobacteria, however, population growth rates did not differ significantly between genotypes from the two time periods. Thus, the costs of resistance are presumably contingent, i.e., dependent on the environment in which the trait is expressed[18], as they apparently become relevant only on the cyanobacterial diet. This suggests that the costs associated with maintaining this trait decreased along with the abundance of cyanobacteria in the lake and that, once the cyanobacteria disappeared, the population no longer experienced costs linked to its adaptation history. Assuming that these costs were contingent implies that they were related specifically to the processing of cyanobacterial food, e.g., to the upregulation of detoxification mechanisms or lipid assimilation efficiencies. During peak-eutrophic times, the costs associated with maintaining the trait were presumably compensated for by the fitness benefits of the trait, i.e., the capacity of coping with dietary cyanobacteria. Decreasing costs with oligotrophication imply that there were additional selection forces involved ultimately favoring the loss of resistance. Although our data do not indicate what these costs may have been, one possibility is that the cost of tolerance is only expressed in the presence of some other physiological stress[35]. Because with oligotrophication phytoplankton in Lake Constance became less dense[36] and would have become increasingly phosphorus limited[37], costs associated with tolerance of cyanobacteria existed under oligotrophic conditions that were not expressed in our experiment with high density high-phosphorus *Acutodesmus* used as good food. We know, for example, that for *D. galeata* from Lake Constance, the combination of food scarcity and low-P content has a large effect on somatic growth rate[37]. Consistent with this decline in phytoplankton abundance and P content, the abundance of *D. galeata* in Lake Constance decreased with oligotrophication[38]. In recent years, *D. galeata* has almost completely disappeared from Lake Constance[39], presumably because it is less competitive than the co-occurring species *D. longispina* under oligotrophic conditions[37].

**Concluding remarks**. Ecosystems around the world are suffering from human impact. Efforts to reverse anthropogenic disturbances are steadily growing, yet we lack knowledge of how organisms and whole ecosystems respond to the relaxation of anthropogenic stressors and if they can return to their pre-disturbance state. We studied the consequences of reversing nutrient pollution for the keystone grazer *Daphnia* in Lake Constance. Resurrection of genotypes from the sediment egg bank allowed us to reconstruct temporal changes in the capacity of these animals to cope with cyanobacterial food. The trait 'resistance to cyanobacteria' that evolved during eutrophication was subsequently lost following oligotrophication, leaving genotypes that are highly susceptible to dietary cyanobacteria. Trait evolution and subsequent trait loss progressed at high evolutionary rates, occurring within only a few generations. Ancestral phenotypic plasticity and evolution of plasticity were identified as important eco-evolutionary components mediating the evolution and subsequent loss of grazer resistance, though the relative contribution of the different components to the two opposing selection processes differed. Disentangling the evolutionary components underlying environmental transitions is crucial for understanding how organisms adapt to human-caused environmental change. The use of resurrected decades-old genotypes from the sediment egg bank of Lake Constance revealed a rare example of human-caused rapid evolution and subsequent loss of a fitness-relevant trait in a natural population, highlighting that evolution can be crucially shaped by human action on the scale of few decades[19]. These findings contribute to our understanding of eco-evolutionary dynamics and the capacity of species to adapt to human-caused environmental perturbations, and may help to assess the risk of extinction and biodiversity loss[40] in response to anthropogenic impacts.

## Methods

**Study site**. Upper Lake Constance is a large (surface area 473 km²) and deep (max. depth 252 m) pre-Alpine lake in Central Europe with a distinct eutrophication and re-oligotrophication history. Total phosphorus concentrations during winter mixing ($TP_{mix}$) increased more than tenfold from about 7 µg P L⁻¹ in the 1950s to more than 80 µg P L⁻¹ in the early 1980s[41]. These changes in trophic state also led to shifts in the phytoplankton community composition, with a marked increase in the relative abundance of cyanobacteria (up to 25% of total phytoplankton biomass in summer)[14,42]. Since then, due to a substantial reduction in external phosphorus inputs, total phosphorus concentrations decreased again to levels that are currently similar to those measured in the 1950s[14,43]. This oligotrophication process was again accompanied by changes in the phytoplankton community composition, most notably by a decrease in the abundance of cyanobacteria, which at present comprise less than 5% of total phytoplankton biomass[15,43].

*Daphnia longispina* (formerly *D. hyalina*) was the only *Daphnia* species occurring in Upper Lake Constance before eutrophication. In the late 1950s, with the onset of eutrophication, *Daphnia galeata* invaded the lake and intermittently became the dominant *Daphnia* species[27,38]. With subsequent oligotrophication, the abundance of *D. galeata* steadily decreased again and currently reaches abundances that are only slightly above detection level[39]. A third species, *Daphnia cucullata*, invaded Upper Lake Constance in about 2014 and recently became the dominant *Daphnia* species.

**Resurrection of *Daphnia galeata***. The experiment was performed with *D. galeata* genotypes hatched ("resurrected"[11]) from the sediment 'egg bank'. The extent to which egg banks can be used to reconstruct the microevolutionary history of species producing dormant stages depends on genotype-specific differences in diapausing egg production, the number of offspring hatching from the deposited eggs, and the level of disturbance of the sediment layers affecting the vertical structure of the genetic archive[44–46]. In the absence of this information, like others, we make the simplifying assumption that genotypes do not differ in these respects[11–13,38,47,48]. The clearly varved sediment cores used here were taken from a water depth of 185 m in an area showing no signs of sediment disturbance. Therefore, natural hatching stimuli, such as increasing day length, light exposure, and temperature[45,49], were most likely absent and the genetic archive is thus expected to reflect genotypes of historically occurring planktonic populations. Although the change in abundance of diapause stages over time does not necessarily reflect the short-term success of clonal lineages at a given time period[50,51], the relative abundance of genotypes derived from diapausing egg banks should reveal information about lineages which successfully reproduced sexually and thus contributed genes to the next generation. Our study covers time periods that were also assessed in previous work[12], i.e., we resurrected different genotypes from the same time span. Our results confirm that genotypes from the 1980s were more

resistant to a cyanobacteria-containing diet. We are therefore confident that the potential bias of differential survival of clones preserved in the egg bank is unlikely to have influenced our results. The same applies to the time span diapausing eggs lay dormant. While the oldest genotypes (1960s) from the first study[12] performed worst, the oldest genotypes (1980s) from this study performed best when exposed to dietary cyanobacteria. In both studies, the oldest genotypes resurrected lay dormant for about 40 years.

Sediment cores were taken in spring 2017 in the Bay of Friedrichshafen in Upper Lake Constance (47°36′26.51"N 9°27′45.82"E) using a multicorer[52]. A total of 21 sediment cores were taken; three sediment cores at a time. The cores were split vertically into halves using a metal sheet and stored at 4 °C in the dark. Sediment cores were partitioned into centimeter sections, reflecting the time period from 1945 to 2010. The age of the different sediment layers was determined by counting annual varves (laminations) and distinct sediment layers deposited during major known historic flood events, both of which have been dated previously using radiometric methods[53]. The sediment samples were sieved through a 250 µm mesh and ephippia (protective chitinous shells normally containing two diapausing eggs) were isolated using a stereomicroscope. Ephippia were directly transferred into 24-well-plates containing filtered water (<0.2 µm) from Lake Constance and exposed to artificial daylight (16:8 L:D) and room temperature (20 °C) to stimulate egg hatching. Plates were checked every other day for hatched neonates and the water in the wells was renewed. The oldest diapausing eggs from which viable offspring could be hatched were from 1975. To extend the time span for our analyses we combined our data with those from Hairston et al.[12,13], who also were not able to hatch genotypes from Lake Constance diapausing eggs older than approximately 40 years. Parthenogenetic lines of each genotype were maintained for 6 to 10 generations in 2-L jars at 20 °C in filtered lake water (<0.2 µm) containing 2 mg C L$^{-1}$ of the green alga *Acutodesmus obliquus* (formerly referred to as *Scenedesmus obliquus*; SAG 276-3a).

Eight genotypes from peak-eutrophic times (1975–1985) and nine genotypes from re-oligotrophic times (1999–2008) were used for the experiment. More precisely, the peak-eutrophic time period was represented by four genotypes from 1978–1980 and four genotypes from 1983 to 1985 and the re-oligotrophic time period was represented by three genotypes from 1999–2002, four genotypes from 2004 to 2005, and two genotypes from 2006–2008.

The species identity of the isolated genotypes was determined based on a panel of ten microsatellite loci[54]. Primers were multiplexed (SwiD15, SwiD10, SwiD1, and 10_14 at 0.3 µM; SwiD4, SwiD14, SwiD2[55], and SwiD5 at 0.5 µM; Dp512 at 0.5 µM; SwiD12 at 0.6 µM) for a single PCR reaction with 2.8 µL of pure water, 5.5 µL of Multiplex PCR Master Mix (Qiagen, Hilden, Germany), 1.7 µM of the multiplexed primers, and 1 µL of DNA. The PCR (Primary denaturation: 95 °C for 15 min; Cycle: 94 °C for 0.5 min, 54 °C for 1.5 min, 72 °C for 1 min; final extension: 60 °C for 30 min) was run for 32 cycles and the products were sequenced on an ABI 3130 XL Sequencer (Applied Biosystems). Microsatellites were scored using STRand (v 2.4.110) and species identity was assigned with the reference panel from Alric et al.[54] using a factorial correspondence analysis, as implemented in Genetix 4.05[56].

**Life-history experiment.** Clonal resistance to cyanobacteria was assessed using two food environments, good and poor food, according to Hairston et al.[12,13]. The poor food consisted of a mixture of the toxic cyanobacterium *Microcystis aeruginosa* (PCC 7806; 20% of total provided carbon) and the green alga *Acutodesmus obliquus* (SAG 276-3a; 80% of total provided carbon). During the years of maximum eutrophication, cyanobacteria comprised about 20% of total summer phytoplankton biomass[13]. The good food consisted of 100% *A. obliquus*. *M. aeruginosa* and *A. obliquus* were each cultured semi-continuously in Cyano medium[57] at 20 °C, illumination at 120 and 60 µmol m$^{-2}$ s$^{-1}$, respectively, and a dilution rate of 0.25 d$^{-1}$ in aerated 5 L vessels. Food suspensions were obtained by centrifugation and resuspension in filtered (<0.2 µm) lake water. The carbon concentrations of the food suspensions were estimated from photometric light extinction (480 nm) and from carbon-extinction equations determined previously.

The experiment was conducted with third-clutch offspring born within 12 h. Neonates were placed in jars containing 80 mL of filtered (<0.2 µm) lake water and 2 mg C L$^{-1}$ of the respective food. A subsample of 20 neonates was taken from the isolated cohort of each genotype for initial dry mass determination after freeze-drying. Initially, each treatment consisted of 10 jars each containing two neonates. *Daphnia* were transferred every second day into new jars with freshly prepared food suspensions. At day five, one individual was taken out of each jar, rinsed with ultra-pure water, transferred into pre-weighed aluminum boats, and stored at −80 °C until they were freeze-dried and weighed (±0.1 µg) for dry mass determination.

**Data analysis.** Mass-specific juvenile somatic growth rates ($g$) were determined as the increase in dry mass from the beginning ($M_0$) until day five of the experiment ($M_t$) with time ($t$) expressed as age in days:

$$g = \frac{\ln(M_t) - \ln(M_0)}{t} \qquad (1)$$

The remaining daphnids (10 replicates per treatment, per genotype) were kept in their respective treatments until they released their third-clutch offspring. The

number of viable offspring produced in each reproduction cycle was determined and population growth rates ($r$) were estimated iteratively using the Lotka–Euler equation:

$$1 = \sum_{x=0}^{n} l_x m_x e^{-rx} \qquad (2)$$

where $l_x$ is the age-specific survivorship, $m_x$ is the number of offspring at each reproduction cycle, and $x$ is the age at reproduction in days. Mean values were used for each of the three parameters and genotypes from the different time periods were treated as replicates. For genotypes that were unable to reproduce on the cyanobacteria-containing diet, population growth rates were estimated using the equation:

$$r = b - d \qquad (3)$$

where $b$ is the birth rate (which is 0 in this case) and $d$ is the *per capita* death rate. The *per capita* death rates were derived from the survival data; they represent the slope of the (log) fraction surviving vs. time relationships.

The reduction in somatic growth rates (GRR) caused by the exposure to cyanobacteria was determined according to Hairston et al.[12] using the equation

$$\text{GRR} = \frac{(g_{\text{good}} - g_{\text{poor}})}{g_{\text{good}}} \qquad (4)$$

where $g_{\text{good}}$ and $g_{\text{poor}}$ are the juvenile somatic growth rates ($d^{-1}$) obtained in the respective food environments. Juvenile somatic growth rates ($g$, $d^{-1}$), growth rate reduction (GRR), and reaction norms were analyzed using one-way ANOVAs followed by Tukey's HSD post hoc tests (SigmaPlot, Systat Software, Version 14.0.).

To compare the evolution and loss of resistance to cyanobacteria, and to account for measurements of $g$ made in different laboratories decades apart, we standardized our data to those from Hairston et al.[12,13]. For each food environment, a standardization factor was determined using the equation:

$$\frac{g_{\text{Hairston}}}{g_{\text{Isanta-Navarro}}} \qquad (5)$$

The two data sets were standardized only for graphical visualization (Fig. 1a, c), statistical analyses were performed exclusively on the newly recorded data: non-standardized versions of graphs are provided as supplementary material (Fig. S1).

**Statistics and reproducibility.** The experiment with the above-described set-up was conducted once. Preliminary experiments with a lower number of genotypes revealed similar results.

**Reporting summary.** Further information on research design is available in the Nature Research Reporting Summary linked to this article.

## Data availability

The source data underlying Figs. 1, 2, S1, S2 and S3 are provided as a Source Data file. Other experimental data that support the findings of this study are available from the corresponding author upon reasonable request. Source data are provided with this paper.

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

## Acknowledgements
The authors cordially thank Max Cassell for experimental support and Jelena Pantel, Lutz Becks, and Dieter Ebert for valuable feedback and criticism that greatly improved the manuscript. Martin Wessels and his team are acknowledged for technical support during sampling of sediment cores and Piet Spaak for support with the microsatellite analysis. This work was funded by the Deutsche Forschungsgemeinschaft (DFG, German Research Foundation, 298726046/GRK2272, to D.M.C., D.S., A.M.), the US National Science Foundation (INT-9603204 to N.G.H.), the Austrian Science Fund (FWF): P29667-B25 (to M.M.) and the grant "SeeWandel: Life in Lake Constance—the past, present, and future" within the framework of the Interreg V programme "Alpenrhein-Bodensee-Hochrhein (Germany/Austria/Switzerland/Liechtenstein)", which funds are provided by the European Regional Development Fund as well as the Swiss Confederation and cantons (to M.M. and D.S.). The funders had no role in study design, data collection, and analysis, decision to publish, or preparation of the manuscript.

## Author contributions
J.I.N. and D.M.C. designed the study. J.I.N. performed the experiment and analyzed the data. J.B. characterized the *Daphnia* genotypes genetically. J.I.N. and D.M.C. prepared the manuscript with contributions from N.H., J.B., A.M., D.S and M.M.

## Funding

## Competing interests

The authors declare no competing interests.
