## [Peer Review File · Nature Communications]

REVIEWER COMMENTS

Reviewer #1 (Remarks to the Author):

This manuscript is a sequel to the well-recognised studies in *Nature* (1999) and *Evolution* (2001), led by one of the authors of the present manuscript, focussing on how rapid evolution made grazers more tolerant to cyanobacterial toxins as exposure to the toxins eventually increased. This was, 20 years ago, elegantly shown by resurrecting grazer resting eggs from different time periods. The main message from the present manuscript is that this tolerance to cyanobacterial toxins is lost as the lake becomes less eutrophic, and thereby contains less toxic cyanobacteria, i.e. the return to pre-eutrophication traits among grazers. Implicitly this also suggests that keeping up tolerance to the toxin is costly and therefore rapidly lost when not exposed to the toxin. The data obtained on grazer (*Daphnia galeata*) trait changes spans over almost half a century offering a rare opportunity to address long-term eco-evolutionary interactions. The methods seem well planned and performed, the manuscript is well written (although sometimes becomes somewhat blurred with details) and the results and conclusions are interesting. However, the basic method, as well as the approach and some of the data, are similar to the original papers (see above) which may put the novelty of the present study into question.

General comments

The approach and main conclusions are straight forward although somewhat hidden behind a multitude of (18!) small, colourful sub-figures (including those in the supplementary). As the main conclusions are easy to grasp, I strongly recommend the authors to reduce the number of (sub) figures. Those different aspects may be interesting (and useful) for a more specific audience, rather than for the general reader of this journal.

Minor comments

Line 110 and 115. It is stated that *D. galeata* does not overwinter in the water column, but a few lines further down (115) the probability is raised that they may have been present in the water column for decades. Please clarify this.

Line 178. "... the genotypes that invaded Lake Constance in the late 1950s". Please clarify here why you state that the genotype(s?) invaded the lake in the 1950ties, e.g. from where? Was it a documented invasion? Why was such a large lake suddenly invaded by genotypes that then became dominant in the lake?

Line 206. *D. galeata* vs *D. longispina*. It would have been a novel touch of this study to assess the GRR also for *D. longispina* in a similar way as for *D. galeata*. A suitable hypothesis would then be that *D. longispina* is far more sensitive to the cyanobacteria than *D. galeata*, i.e. has a far larger GRR. Such a study (which may be easily performed at least on present *longispina*) would indeed broaden the context of the study by including competitive advantages with a "tolerance trait" at different states of a lake (eutrophic vs oligotrophic). If this aspect is not taken into account I strongly recommend to omit lines 204-208.

Line 269. Some more information on the methods behind the dating of the sediment layers are needed. Not enough to just provide a reference.

Reviewer #2 (Remarks to the Author):

The authors follow up on an earlier pair of papers that described how *Daphnia galeata* adapted to the eutrifying environment of Lake Constance from 1960-1980. The water quality has since returned to its oligotrophic past and they demonstrate that the *Daphnia* have evolved accordingly. This work is

enabled by the durability of the sexual eggs produced by this species at the end of the summer. Unhatched eggs accumulate in the sediment at the bottom of the lake and can be mined and approximately dated from cores collected from the lake bottom. The eggs are isolated, hatched in the laboratory, cultured for three generations, then evaluated for the juvenile somatic growth rate and population growth rate when reared on diets that represent either the eutrophic or oligotrophic phases of the lake. Cyanobacteria become a dominant component of the phytoplankton in eutrophic conditions and are a poorer source of nutrition.

D. galeata is an invasive species that appeared in the lake at around the time that eutrophication began, then increased in abundance and apparently displaced *D. longispina*, in the process. The earlier papers compared *D. galeata* clones from their first appearance in the lake with ones collected after eutrophication. Clones derived from the eutrophicated conditions had higher juvenile and population growth rates on a diet containing cyanobacteria than those from the time period before the lake became eutrophicated. The current study shows that the reverse occurred between the period of peak eutrophication in the 1980's and 2008. They also showed that the earlier evolution was characterized by counter-gradient selection – the juvenile growth rate on a diet with cyanobacteria was associated with accelerated growth on a diet without the cyanobacteria. Their analysis shows that the change in the mean phenotype of the population is consistent with evolution, as opposed to drift in the absence of selection in favor of good performance on a diet with cyanobacteria, implying that there was a cost associated with adaptation to a diet rich in cyanobacteria. The analysis also includes a decomposition of the contributions of ancestral plasticity, the evolution of plasticity and genetic change to each phase of adaptation. The adaptation of *D. galeata* to an oligotrophic environment was accompanied by a substantial decline in its abundance and an increase in the abundance of *D. longispina*.

The authors used anthropogenic environmental change as a conceptual framework for the results. This is a popular, and sometimes overused, theme but one that is highly relevant here because of the way the work tightly links human caused changes to the environment with ecology (species replacement) and the underlying evolution. The role of evolution in response to anthropogenic change is not a new topic, but it is an important contribution to our understanding of the way anthropogenic change affects ecology, then the evolution of the species that interact in natural ecosystems.

I have only a few suggestions for revision. First, one limitation of such work, and a mild limitation in my view, is that the results could be influenced by the differential survival of clones preserved in the egg bank. The fact that the current study includes time periods that were part of the original study should give them the ability to address this issue and to show that such a bias is unlikely to have influenced the results. I also found the "Reversed selection on a costly trait" paragraph (lines 190-208) confusing. *D. galeata* adapted to a diet of cyanobacteria grow faster than those from oligotrophic environments on a diet containing cyanobacteria, but also grow as well on a diet without cyanobacteria (hence their invoking counter gradient selection). On lines 197 – 200, they argue that this means that the costs associated with this adaptation had been compensated for, yet on lines 190 – 197 they argue that there are such costs because genotypes from the most recent samples are more adversely affected by a diet with cyanobacteria. If you put these different observations together, it says that genotypes adapted to cyanobacteria are equal in oligotrophic environments and better in eutrophic environments. If so, why did *D. galeata* evolve when the lakewaters were cleaned up after 1980? It must be that the *D. galeata* from 1999 - 2008 have some advantage over those from the 1980's that was not represented in these assays.

Reviewer 1:

This manuscript is a sequel to the well-recognised studies in Nature (1999) and Evolution (2001), led by one of the authors of the present manuscript, focussing on how rapid evolution made grazers more tolerant to cyanobacterial toxins as exposure to the toxins eventually increased. This was, 20 years ago, elegantly shown by resurrecting grazer resting eggs from different time periods. The main message from the present manuscript is that this tolerance to cyanobacterial toxins is lost as the lake becomes less eutrophic, and thereby contains less toxic cyanobacteria, i.e. the return to pre-eutrophication traits among grazers. Implicitly this also suggests that keeping up tolerance to the toxin is costly and therefore rapidly lost when not exposed to the toxin. The data obtained on grazer (*Daphnia galeata*) trait changes spans over almost half a century offering a rare opportunity to address long-term eco-evolutionary interactions. The methods seem well planned and performed, the manuscript is well written (although sometimes becomes somewhat blurred with details) and the results and conclusions are interesting. However, the basic method, as well as the approach and some of the data, are similar to the original papers (see above) which may put the novelty of the present study into question.

Response: It is correct that our manuscript builds on the studies that have been performed more than 20 years ago. We believe that this is an advantage rather than a limitation and while the method itself might not be entirely novel, the outcome of our study surely is. We are presenting a unique data set that covers data for more than half a century in which Lake Constance underwent drastic anthropogenic changes. Currently, efforts to reverse anthropogenic impacts on our ecosystems are intensified. One central assumption is that once we reverse the anthropogenic impact, the ecosystem falls back into its pre-disturbed state. However, we know little about how the organisms react evolutionarily to reversed anthropogenic change. There are very few – if any – examples of such a clear and well-studied system where the entire process of pre-disturbed, disturbed and post-disturbed conditions has been observed and studied. None of those examples has addressed a freshwater ecosystem, hence we believe the complete data set that we present here has the potential of becoming a classic text-book example.

General comments

The approach and main conclusions are straight forward although somewhat hidden behind a multitude of (18!) small, colourful sub-figures (including those in the supplementary). As the main conclusions are easy to grasp, I strongly recommend the authors to reduce the number of (sub) figures. Those different aspects may be interesting (and useful) for a more specific audience, rather than for the general reader of this journal.

Response: We reduced the number of sub-figures in the main text from 10 to 8 (presented in two figures) and the number of figures in the supplements to 3 (5 sub-figures). By removing former Figure 3b (survival rates) and former Figure 2 c+d and S4 we reduced the number of different aspects presented in the manuscript because we agree this will be more interesting for a more specific audience. Furthermore, we reduced the content in Figure 1b and 1d.

Minor comments

Line 110 and 115. It is stated that *D. galeata* does not overwinter in the water column,

but a few lines further down (115) the probability is raised that they may have been present in the water column for decades. Please clarify this.

Response: To clarify this, we rephrased this section. It now reads: “D. galeata abundance is below detection level in the water column of Lake Constance during winter, which strongly suggests that annual recruitment from the sediment egg bank is crucial for population persistence.”

Line 178. “... the genotypes that invaded Lake Constance in the late 1950s”. Please clarify here why you state that the genotype(s?) invaded the lake in the 1950ties, e.g. from where? Was it a documented invasion? Why was such a large lake suddenly invaded by genotypes that then became dominant in the lake?

Response: The invasion of Daphnia galeata is a documented invasion (Straile, D., & Geller, W. (1998). Crustacean zooplankton in Lake Constance from 1920 to 1995: Response to eutrophication and re-oligotrophication. Advances in Limnology, 53, 255–274.) that we now cite in the text. Why such a large lake like Lake Constance was suddenly invaded by this species remains unknown; knowing this would certainly provide highly relevant insights. However, general patterns in peri-alpine lakes suggest that changes in trophic conditions are likely to have facilitated the establishment of Daphnia galeata (Brede, N. et al. (2009). The impact of human-made ecological changes on the genetic architecture of Daphnia species. Proc. Natl. Acad. Sci. 106, 4758–4763).

Line 206. D. galeata vs D. longispina. It would have been a novel touch of this study to assess the GRR also for D longispina in a similar way as for D. galeata. A suitable hypothesis would then be that D. longispina is far more sensitive to the cyanobacteria than D. galeata, i.e. has a far larger GRR, Such a study (which may be easily performed at least on present longispina) would indeed broaden the context of the study by including competitive advantages with a “tolerance trait” at different states of a lake (eutrophic vs oligotrophic). If this aspect is not taken into account I strongly recommend to omit lines 204-208.

Response: We agree that this would have been interesting, unfortunately this has been found to be methodologically impossible. Neither the previous study on this D. galeata population (Hairston et al, 1999) nor our study, conducted more than 20 years later, was able to resurrect Daphnia longispina genotypes from sediment cores. Resting eggs of D. longispina can be found in the sediment of Lake Constance (Brede et al, 2009), but they do not hatch. What prevents those eggs from hatching is not known. Comparing decades-old D. galeata genotypes resurrected from the sediment with recent D. longispina genotypes isolated from the water column, would not have given any useful insights. Having D. longispina genotypes from the same different trophic states that Lake Constance underwent would have surely been valuable. What would have been possible, is to isolate D. longispina directly from the water column and compare them to D. galeata genotypes also isolated from the water column. However, this comparison of recent D. longispina vs. D. galeata has already been done, even for Lake Constance, in a study that we cited already in our manuscript (Spaak et al, 2012). The authors of that study clearly show, for several lakes, that D. galeata is dominating in eutrophic and D. longispina in oligotrophic lakes. We believe that this is an important information for the readers and therefore disagree to omit those lines.

Line 269. Some more information on the methods behind the dating of the sediment layers are needed. Not enough to just provide a reference.

Response: We now provide more details in addition to the reference. The section now reads: "The age of the different sediment layers was determined by counting annual varves (laminations) and distinct sediment layers deposited during major known historic flood events, both of which have been dated previously using radiometric methods."

Reviewer 2:

The authors follow up on an earlier pair of papers that described how *Daphnia galeata* adapted to the eutriching environment of Lake Constance from 1960-1980. The water quality has since returned to its oligotrophic past and they demonstrate that the *Daphnia* have evolved accordingly. This work is enabled by the durability of the sexual eggs produced by this species at the end of the summer. Unhatched eggs accumulate in the sediment at the bottom of the lake and can be mined and approximately dated from cores collected from the lake bottom. The eggs are isolated, hatched in the laboratory, cultured for three generations, then evaluated for the juvenile somatic growth rate and population growth rate when reared on diets that represent either the eutrophic or oligotrophic phases of the lake. Cyanobacteria become a dominant component of the phytoplankton in eutrophic conditions and are a poorer source of nutrition. *D. galeata* is an invasive species that appeared in the lake at around the time that eutrophication began, then increased in abundance and apparently displaced *D. longispina*, in the process. The earlier papers compared *D. galeata* clones from their first appearance in the lake with ones collected after eutrichification. Clones derived from the eutrichified conditions had higher juvenile and population growth rates on a diet containing cyanobacteria than those from the time period before the lake became eutrichified. The current study shows that the reverse occurred between the period of peak eutrichification in the 1980's and 2008. They also showed that the earlier evolution was characterized by counter-gradient selection – the juvenile growth rate on a diet with cyanobacteria was associated with accelerated growth on a diet without the cyanobacteria. Their analysis shows that the change in the mean phenotype of the population is consistent with evolution, as opposed to drift in the absence of selection in favor of good performance on a diet with cyanobacteria, implying that there was a cost associated with adaptation to a diet rich in cyanobacteria. The analysis also includes a decomposition of the contributions of ancestral plasticity, the evolution of plasticity and genetic change to each phase of adaptation. The adaptation of *D. galeata* to an oligotrophic environment was accompanied by a substantial decline in its abundance and an increase in the abundance of *D. longispina*.

The authors used anthropogenic environmental change as a conceptual framework for the results. This is a popular, and sometimes overused, theme but one that is highly relevant here because of the way the work tightly links human caused changes to the environment with ecology (species replacement) and the underlying evolution. The role of evolution in response to anthropogenic change is not a new topic, but it is an important contribution to our understanding of the way anthropogenic change affects ecology, then the evolution of the species that interact in natural ecosystems.

Response: We are pleased by these comments.

I have only a few suggestions for revision. First, one limitation of such work, and a mild limitation in my view, is that the results could be influenced by the differential survival of clones preserved in the egg bank. The fact that the current study includes time periods that were part of the original study should give them the ability to address this issue and to show that such a bias is unlikely to have influenced the results.

Response: We added some more sentences to this paragraph (ll. 267- 273) that read: "Our study covers time periods that were also assessed in previous work¹², i.e., we resurrected different genotypes from the same time span. Our results confirm that genotypes from the 1980s were resistant to a cyanobacteria containing diet. We are therefore confident that the potential bias of differential survival of clones preserved in the egg bank is unlikely

to have influenced our results. The same applies to the time span diapausing eggs lay dormant. While the oldest genotypes (1960s) from the first study¹² performed worst, the oldest genotypes (1980s) from this study, performed best when exposed to dietary cyanobacteria. In both studies, the oldest genotypes resurrected lay dormant for about 40 years.”

I also found the “Reversed selection on a costly trait” paragraph (lines 190-208) confusing. *D. galeata* adapted to a diet of cyanobacteria grow faster than those from oligotrophic environments on a diet containing cyanobacteria, but also grow as well on a diet without cyanobacteria (hence their invoking counter gradient selection). On lines 197 – 200, they argue that this means that the costs associated with this adaptation had been compensated for, yet on lines 190 – 197 they argue that there are such costs because genotypes from the most recent samples are more adversely affected by a diet with cyanobacteria. If you put these different observations together, it says that genotypes adapted to cyanobacteria are equal in oligotrophic environments and better in eutrophic environments. If so, why did *D. galeata* evolve when the lakewaters were cleaned up after 1980? It must be that the *D. galeata* from 1999 - 2008 have some advantage over those from the 1980’s that was not represented in these assays.

Response: We agree with this conclusion and have rewritten that paragraph to resolve the confusion.

REVIEWERS' COMMENTS

Reviewer #1 (Remarks to the Author):

Although I'm still not convinced by the author's response regarding novelty (a "unique" data set is not the same as novelty since all data sets can, in some way or another, be considered as unique), I strongly agree that the conclusions from the ms. are very interesting and important. Moreover, the study is well performed and covers a long time period and I also agree that the paper may, just as its original studies by one of the co-authors on the same system, become a text-book example of how anthropogenic effects may drive evolution in different directions. The authors have made some modifications according to my suggestions and I have no further comments.

Reviewer #2 (Remarks to the Author):

I thought the original manuscript was well written and exciting and only offered two suggestions for revision. The authors dealt with both of them to my satisfaction. I saw no issues with the comments offered by the other reviewer. I feel that this paper is ready for publication.

Reviewer 1:

Although I'm still not convinced by the author's response regarding novelty (a "unique" data set is not the same as novelty since all data sets can, in some way or another, be considered as unique), I strongly agree that the conclusions from the ms. are very interesting and important. Moreover, the study is well performed and covers a long time period and I also agree that the paper may, just as its original studies by one of the co-authors on the same system, become a text-book example of how anthropogenic effects may drive evolution in different directions. The authors have made some modifications according to my suggestions and I have no further comments.

Response: Thank you for your suggestions, which further improved the quality of our manuscript.

Reviewer 2:

I thought the original manuscript was well written and exciting and only offered two suggestions for revision. The authors dealt with both of them to my satisfaction. I saw no issues with the comments offered by the other reviewer. I feel that this paper is ready for publication.

Response: Thank you for your suggestions, which further improved the quality of our manuscript.